# Serum chemerin levels: A potential biomarker of joint inflammation in women with rheumatoid arthritis

**Fabiola Gonzalez-Ponce**[1‡], **Jorge I. Gamez-Nava**[1,2,3‡], **Emilio E. Perez-Guerrero**[4], **Ana M. Saldaña-Cruz**[1], **Maria L. Vazquez-Villegas**[2,5], **Juan M. Ponce-Guarneros**[1,6], **Miguel Huerta**[7], **Xochitl Trujillo**[7], **Betsabe Contreras-Haro**[8], **Alberto D. Rocha-Muñoz**[8], **Maria O. Carrillo-Escalante**[2], **Esther N. Sanchez-Rodriguez**[1], **Eli E. Gomez-Ramirez**[1], **Cesar A. Nava-Valdivia**[9], **Ernesto G. Cardona-Muñoz**[1]*, **Laura Gonzalez-Lopez**[1,2,10]*, **on behalf of the Research Group for the Assessment of Prognosis Biomarkers in Autoimmune Disorders**[¶]

**1** Departamento de Fisiología Centro Universitario de Ciencias de la Salud, Programa de Doctorado en Farmacología, Instituto de Terapeutica Experimental y Clínica, Universidad de Guadalajara, Guadalajara, Jalisco, Mexico, **2** Departamento de Salud Pública, Programa de Doctorado en Salud Publica, Centro Universitario de Ciencias de la Salud, Universidad de Guadalajara, Guadalajara, Jalisco, Mexico, **3** Unidad de Investigación Biomédica 02, Hospital de Especialidades, Centro Médico Nacional de Occidente, Instituto Mexicano del Seguro Social, Guadalajara, Jalisco, Mexico, **4** Instituto de Investigación en Ciencias Biomédicas, Centro Universitario de Ciencias de la Salud, Universidad de Guadalajara, Guadalajara, Jalisco, Mexico, **5** Servicio de Epidemiología, Unidad de Medicina Familiar número 04, Instituto Mexicano del Seguro Social, Guadalajara, Jalisco, México, **6** Unidad Medica Familiar 97, Instituto Mexicano del Seguro Social, Magdalena, Jalisco, México, **7** Centro Universitario de Investigaciones Biomedicas, Universidad de Colima, Colima, Mexico, **8** División de Ciencias de la Salud, Departamento de Ciencias Biomédicas, Departamento Salud-Enfermedad como Proceso Individual, Centro Universitario de Tonalá, Universidad de Guadalajara, Tonalá, Jalisco México, **9** Departamento de Microbiologia y Patologia, Centro Universitario de Ciencias de la Salud, Universidad de Guadalajara, Guadalajara, Jalisco, Mexico, **10** Departamento de Medicina Interna-Reumatología, Hospital General Regional 110, Instituto Mexicano del Seguro Social, Guadalajara, Jalisco, Mexico

‡ FGP and JIGN participated equally in this work and should be considered as co-first authors.
¶ Membership of the Research Group for the Assessment of Prognosis Biomarkers in Autoimmune Disorders can be found in the Acknowledgments.
* lauraacademicoudg@gmail.com, dralauragonzalez@prodigy.net.mx (LGL); cameg1@gmail.com, cardona@cucs.udg.mx (EGCM)

**Data Availability Statement:** All relevant data are within the manuscript.

## Abstract

### Background

Chemerin has a potential role in perpetuating inflammation in autoimmune diseases. Nevertheless, to date, there is no conclusive information on whether high chemerin levels increase the severity of rheumatoid arthritis (RA). Therefore, this study evaluated whether serum chemerin is a biomarker of disease activity in RA patients.

### Methods

Study design: cross-sectional. The assessment included clinical and laboratory characteristics, body mass index (BMI) and fat mass. The severity of the disease activity was identified according to the DAS28-CRP index as follows: A) RA with a DAS28-CRP≤2.9 (remission/mild activity) and B) RA with a DAS28-CRP>2.9 (moderate/severe activity). Serum

**Funding:** The author(s) received no specific funding for this work.

**Competing interests:** The authors have declared that no competing interests exist.

chemerin concentrations were measured by ELISA, and ≥103 ng/mL was considered a high level. Logistic regression analysis was applied to determine whether high chemerin levels were associated with disease activity in RA after adjusting for confounders. Multiple regression analysis was performed to identify variables associated with chemerin levels.

## Results

Of 210 RA patients, 89 (42%) subjects had moderate/severe disease activity and had higher serum chemerin levels than patients with low disease activity or remission (86 ± 34 vs 73± 27; p = 0.003). Serum chemerin correlated with the number of swollen joints (r = 0.15; p = 0.03), DAS28-CRP (r = 0.22; p = 0.002), and C-reactive protein levels (r = 0.14; p = 0.04), but no correlation was observed with BMI and fat mass. In the adjusted logistic regression analysis, high chemerin levels (≥103 ng/mL) were associated with an increased risk of moderate/severe disease activity (OR: 2.76, 95% CI 1.35–5.62; p = 0.005). In the multiple regression analysis, after adjusting for potential confounders, serum chemerin levels were associated with higher DAS28-CRP (p = 0.002).

## Conclusions

Higher chemerin levels increased the risk of moderate and severe disease activity in RA. These results support the role of chemerin as a marker of inflammation in RA. Follow-up studies will identify if maintaining low chemerin levels can be used as a therapeutic target.

## Introduction

Rheumatoid arthritis (RA) is an inflammatory, chronic, progressive disease involving synovial joints and characterized by bone and cartilage erosions associated with a progressive decrease in joint functioning, leading to disability and impaired quality of life [1]. One of the main clinical challenges in RA, is the persistence of joint inflammation in many patients despite treatment with synthetic disease-modified antirheumatic drugs (synthetic DMARDs) or with biologic disease-modified antirheumatic drugs (biologic DMARDs) [2–4].

Several biomarkers of disease activity have been proposed in these patients, of them the most used are C-reactive protein (CRP) or erythrocyte sedimentation rate (ESR) included in most indices of disease activity, including the Disease Activity Score of 28 Joints (DAS28) [3, 5]. Recently, some adipokines were found to be involved in several inflammatory processes lead by the immune system [6]. Among these adipokines, chemerin has been identified as a molecule implicated in the inflammation mediated by immune mechanisms [7]. Chemerin is a chemoattractant adipokine secreted by immature dendritic cells and macrophages that binds to the G-protein coupled receptor CMKLR1 (ChemR23) [8]. Chemerin participates in adipocyte regulation and has proinflammatory activity in endothelial cells [9]. Chemerin stimulates macrophage and dendritic cell adhesion and maturation. Chemerin also has a role in the activation of fibroblast cells and synoviocytes [7]. Some previous studies have described a possible relationship between chemerin and inflammation in RA patients [7, 10]. Ha et al. identified increased chemerin in RA compared with controls, and the severity of the disease activity was correlated with the elevated chemerin levels [11]. Additionally, several published works have identified that chemerin levels can decrease in response to biologic DMARDs [12–14]. Of these studies, Herenius et al. identified a decrease in chemerin in RA patients treated with

adalimumab [12]. Similarly, Makrilakis et al. found that after 6 months of treatment with tocilizumab, there was a decrease in the serum levels of chemerin [13]. Finally, Fioravanti et al. found that although tocilizumab can induce a decrease in chemerin levels, this reduction was not correlated with other parameters of disease activity [14].

Recently, our group have reported the results of a study assessing the relation between chemerin levels and functional disability [15]. In this report performed in a small group of patients with RA, was identified that those subjects with impairment in the functioning had increased serum chemerin levels, mainly among patients who had high disease activity [15]. In persons with noninflammatory rheumatic disorders, chemerin levels are associated with obesity and metabolic syndrome [16]. In early RA patients managed using the treat to target strategy, Tolusso et al. identified that chemerin levels can be considered a predictor of early remission of inflammation [17]; however, other studies are required to validate these findings. These data published in the literature support the hypothesis that chemerin levels are related to disease activity in RA [11, 15, 18]. Nevertheless, there are no previous studies assessing the relation between the persistence of disease activity in RA patients treated with DMARDs and serum chemerin levels.

Therefore, the objective of this study was to evaluate whether serum chemerin is an independent biomarker of moderate or severe disease activity in RA patients.

## Patients and methods

We included 210 women with RA being attended at a secondary-care hospital in Guadalajara, Mexico. They voluntarily agreed to be included in this study.

### Ethics and consent

This observational study was designed following the principles of the 64th Declaration of Helsinki (last revision Fortaleza, Brazil 2013) and the national regulations for research studies in humans. The Research and Ethics Committee of the Hospital General Regional #110 del Instituto Mexicano del Seguro Social in Guadalajara, Mexico, approved this study (code of approval: R-2016-1303-11). All participants in this study signed a voluntary informed consent form before study inclusion.

**Inclusion criteria.** a)>18 years old, b) women and c) met the 1987 American College of Rheumatology (ACR) criteria [19]. All patients were being treated at the time of the study with synthetic DMARDs and/or biologic DMARDs. We excluded patients with a history of ischemic cardiopathy, myocardial infarction and stroke, overlapping syndromes, thyroid disease, chronic renal failure (serum creatinine >1.5 mg/dL), active infections, psoriasis, cancer, or pregnancy. All patients of reproductive age had to be using a contraceptive method. Patients with type 2 diabetes mellitus were allowed to participate in the study if they were taking oral antidiabetics. Patients using insulin were excluded. Patients with hypertension were allowed to participate if they were taking antihypertensive drugs at stable doses and if they had controlled hypertension. Patients with overweight or obesity were allowed to participate, and body mass index was considered a variable to be adjusted in the statistical analysis.

**Study development.** Trained researchers performed a structured review of the epidemiological, clinical, and therapeutic characteristics. Disease activity was assessed using the Disease Activity Score of 28 Joints (DAS28). The DAS28 index includes four components: i) 28 swollen joint counts, ii) 28 tender joint counts, iii) a global health index perceived by the patient, and iv) an acute phase reactant (ESR or CRP) [20, 21]. We used the DAS28-CRP index to identify two groups: A) RA patients with a DAS28-CRP≤2.9 (RA in remission/mild activity) and B) RA patients with a DAS28-CRP>2.9 (RA with moderate/severe activity) [20].

We assessed physical functioning limitations using the Spanish adaptation of the Health Assessment Questionnaire Disability Index (HAQ-DI) [22]. In the HAQ-DI, RA patients self-reported the amount of difficulty perceived when performing daily living activities, reflecting the impairment of functioning in the previous week. A higher score in the HAQ-DI indicates more functional disability [22].

## Determination of Body Mass Index (BMI)

BMI was determined using the Quetelet formula (weight (kg)/height (m)$^2$) [23].

## Determination of fat mass (%)

Fat mass (%) was assessed by trained researchers using dual-energy X-ray absorptiometry (DXA) (LUNAR 2000, Prodigy Advance; General Electric™ equipment, Madison, WI, USA).

## Chemerin level determination

Serum chemerin levels were quantified using an enzyme-linked immunosorbent assay (ELISA) (Quantikine, R&D Systems Human Chemerin Immunoassay). The sensitivity of this assay was 7.80 pg/mL, and the samples were double-checked.

## Other laboratory studies

Rheumatoid factor IU/mL (RF) and C-reactive protein mg/dL (CRP) were quantified using nephelometry. Anti-cyclic citrullinated peptide antibodies (anti-CCP) were determined by ELISA using second-generation anti-CCP (anti-CCP2) RU/mL (Euroimmun, Medizinische Labordiagnostika, Germany). Cutoff points: RF >20 IU/mL; CRP >10 mg/L; anti-CCP2 >5 RU/mL.

## Statistical analysis

Quantitative variables are expressed as the mean ± standard deviation (SD), and qualitative variables as frequencies (%). Student's t-tests were computed to compare means between groups. Chi-square tests were used to compare proportions between groups. Pearson correlation tests were used to identify the strength of the association between chemerin, DAS28-CRP, BMI, fat mass (%) and other quantitative variables. Because there is no established cutoff point of high levels of chemerin, we identified higher levels in our RA population by selecting the cutoff point from levels equal to or above the 80th percentile of our data; then the cutoff for high chemerin levels was considered ≥103 ng/mL.

A multivariate logistic regression model was used to identify variables associated with disease activity. The final model was obtained using forward stepwise analysis with adjusted odds ratios (ORs) and 95% confidence intervals (95% CIs). A multivariate multiple regression analysis was used to test the variables related to chemerin concentrations. All variables with a p value ≤0.1 in the bivariate analysis were included in the multivariate analysis. We used R version 4.0.3 (R Core Team 2020) to perform the statistical analyses. Figures were constructed in R using the ggplot2 package. A p-value ≤0.05 was considered statistically significant.

## Results

Fig 1 shows the study flowchart. Although 264 women with RA being treated at an outpatient clinic were screened for inclusion, we excluded 54 patients for the following reasons: hypothyroidism n = 17, cancer n = 2, overlapping syndromes n = 7, current infection n = 15, serum creatinine >1.5 mg/dL n = 1, psoriasis n = 1, history of ischemic cardiovascular disease n = 4,

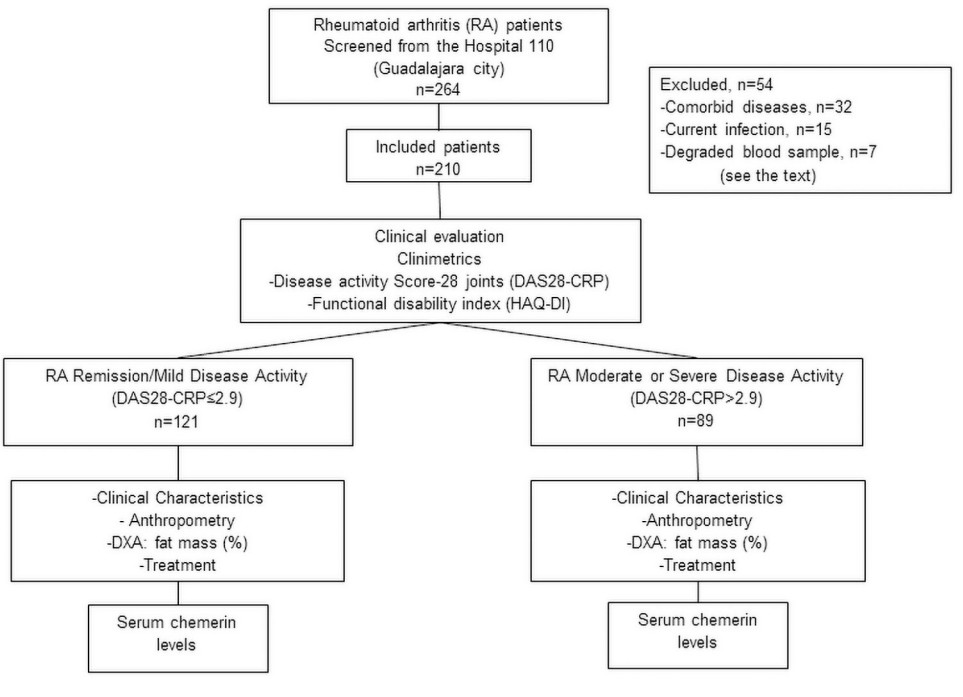

**Fig 1. Study flowchart.**

and inadequate blood sample for the quantification of chemerin n = 7. Consequently, we included 210 women with RA for the analysis. Of the 210 patients with RA, 121 were classified as remission or mild disease activity, and 89 were classified as moderate or severe disease activity.

## Comparison of clinical characteristics between RA in remission/mild activity versus RA with moderate or severe activity

Table 1 describes the characteristics of all patients with RA. The mean age of these women with RA was 56.59 ± 11.25 years and they had a mean disease duration of 12.59 ± 8.58 years. Among these patients, 99% were receiving synthetic DMARDs. Among the total patients who were included, 77% were receiving glucocorticoids. In data that are not shown in the tables, all but one of the patients was receiving synthetic DMARDs: methotrexate (63%), leflunomide (36%), sulfasalazine (31%), chloroquine (14%) or azathioprine (14%). Combination therapy with >1 DMARD was being used by 107 patients (51%). Biologic DMARDs were used by 25 (12%) of the RA patients [etanercept 18 (8.6%), rituximab 5 (2.4%), infliximab 1 (0.5%), and adalimumab 1 (0.5%)]. There were no differences in the medication regimen between the moderate/severe disease activity or mild/remission disease activity groups. Some of the patients used medications for comorbidities; of them, we found that among the patients with diabetes mellitus, 83% (24) used metformin and 17% (5) used glibenclamide; among the patients with high blood pressure, 10% (8) used beta-blockers, 10% (8) calcium blockers, 49% (40) angiotensin converting enzyme inhibitors, and 31% (25) angiotensin II receptor antagonists.

In data that are not shown in tables, we compared the serum chemerin levels between RA patients with and without diabetes mellitus without observing statistical differences between these groups (78.09 ± 29.03 ng/mL vs 83.76 ± 38.60 respectively, p = 0.454). Similarly, the

**Table 1. Comparison of clinical characteristics between rheumatoid arthritis patients with remission or mild activity and those with moderate or severe activity.**

|  | Overall | Disease Activity | | |
| --- | --- | --- | --- | --- |
| Variables | RA | Moderate or severe activity (>2.9) | Remission or mild activity (≤2.9) | p value |
|  | n = 210 | n = 89 | n = 121 |  |
| Age, *mean ± SD* | 56.59 ± 11.25 | 56.54 ± 9.51 | 56.62 ± 12.42 | 0.958 |
| Smoking, n (%) | 18 (9) | 9 (10) | 9 (7) | 0.494 |
| Menopause, n (%) | 164 (78) | 74 (83) | 90 (74) | 0.129 |
| BMI, *mean ± SD* | 27.73 ± 4.35 | 27.96 ± 4.12 | 27.55 ± 4.52 | 0.499 |
| Fat mass (%), *mean ± SD* | 46.33 ± 5.94 | 47.04 ± 5.81 | 45.81 ± 6.00 | 0.141 |
| **Comorbidities** |  |  |  |  |
| Diabetes mellitus, n (%) | 29 (14) | 17 (19) | 12 (10) | 0.057 |
| Hypertension, n (%) | 81 (39) | 37 (42) | 44 (36) | 0.443 |
| Disease duration, years, *mean ± SD* | 13 ± 9 | 13 ± 9 | 12 ± 9 | 0.425 |
| HAQ-DI Score, *mean ± SD* | 0.50 ± 0.56 | 0.90 ± 0.58 | 0.21 ± 0.31 | **<0.001** |
| Functional disability, n (%) | 78 (37) | 63 (71) | 15 (12) | **<0.001** |
| Swollen joints, *mean ± SD* | 1.3 ± 3.3 | 2.90 ± 4.56 | 0.13 ± 0.69 | **<0.001** |
| Painful joints, *mean ± SD* | 2.17 ± 4.77 | 4.93 ± 6.35 | 0.14 ± 0.47 | **<0.001** |
| Severity of pain, *mean ± SD* | 37.04 ± 28.40 | 55.96 ± 24.70 | 23.13 ± 22.29 | **<0.001** |
| **Treatment** |  |  |  |  |
| Glucocorticoids, n (%) | 161 (77) | 67 (75) | 94 (78) | 0.684 |
| Synthetic-DMARDs, n (%) * | 209 (99) | 89 (100) | 120 (99) | NC |
| Biologic-DMARDs, n (%) | 25 (12) | 8 (9) | 17 (14) | 0.263 |
| **Laboratory measurements** |  |  |  |  |
| Rheumatoid Factor (UI) (+), n (%) | 130 (72) | 56 (75) | 74 (70) | 0.474 |
| ESR (mm/Hr), (+), n (%) | 127 (64) | 57 (70) | 70 (60) | 0.128 |
| CRP (mg/L), (+), n (%) | 118 (56) | 55 (62) | 63 (52) | 0.160 |
| Anti-CCP (RU/mL), (+), n (%) | 128 (73) | 50 (69) | 78 (77) | 0.240 |
| Chemerin (ng/mL), *mean ± SD* | 78.88 ± 30.48 | 86.16 ± 33.67 | 73.50 ± 26.80 | 0.003 |

Frequency of data obtained from the patients: RF: 181 patients; ESR: 198 patients; anti-CCP: 175 patients. Other variables were measured at the time of the study in the total number of patients. Abbreviations: DMARDs: Disease-Modifying Antirheumatic Drugs; Anti-CCP: Anti-Cyclic Citrulinated Peptide.

(*) Fisher's exact test.

NC = not calculated.

presence of diabetes mellitus was compared between the RA group with high levels of chemerin (≥103 ng/mL) versus RA group with normal levels of chemerin (13% vs 18% respectively, p = 0.452).

In addition, Table 1 shows the comparison between clinical characteristics of patients with remission/mild activity and those with moderate/severe activity despite treatment with biologic DMARDs or synthetic DMARDs. Compared to women with mild disease activity or remission, in women with moderate or severe disease activity, a higher HAQ-DI score (p<0.001) and elevated serum chemerin levels (p = 0.003) were identified. Women with moderate or severe disease activity had a higher frequency of functional disability (71%) (p<0.001).

## Correlation between chemerin levels and clinical variables

Table 2 shows the correlations between serum chemerin levels and clinical variables. A positive correlation was found between chemerin and DAS 28-CRP (p = 0.002), CRP (p = 0.039), and swollen joints counts (p = 0.029). No other correlations were found between chemerin levels and any other variables.

**Table 2. Correlation between chemerin levels and clinical variables.**

| Variables | Chemerin | |
|---|---|---|
| | **r** | **p** |
| Age | 0.017 | 0.809 |
| Disease duration | -0.108 | 0.120 |
| Body mass index (kg/m$^2$) | 0.043 | 0.535 |
| Fat mass (%) | 0.098 | 0.158 |
| Swollen joints count | 0.150 | **0.029** |
| Painful joints count | 0.119 | 0.087 |
| Severity of pain | 0.127 | 0.065 |
| Disease Activity Score (DAS 28-CRP) | 0.215 | **0.002** |
| C-reactive protein (mg/dL) | 0.142 | **0.039** |
| Rheumatoid factor (UI) | 0.049 | 0.516 |
| Erythrocyte sedimentation rate (mm/Hr) | 0.067 | 0.349 |
| Anti-CCP (RU/mL) | -0.039 | 0.604 |

Correlations were obtained using the Pearson correlation test. Abbreviations: Anti-CCP: Anti-Cyclic Citrullinated Peptide.

Fig 2 shows the comparisons of chemerin concentrations between the group with DAS28-CRP>2.9 versus DAS28-CRP≤2.9. RA patients with moderate/severe disease activity had higher chemerin levels compared to RA in remission/mild disease activity (p = 0.003).

Comparisons of means between groups were performed using Student t-tests. A p-value of ≤ 0.05 was considered significant.

## Factors associated with moderate/severe disease activity in patients with rheumatoid arthritis

Table 3 shows the results of a logistic regression analysis evaluating the risk factors for moderate or severe disease activity. In RA patients with high chemerin levels (≥103 ng/mL), the risk of moderate/severe disease activity was increased (OR: 2.76, 95% CI 1.35–5.62).

## Factors associated with chemerin serum levels in patients with rheumatoid arthritis

Additionally, in data that are not shown in the tables, we performed a multiple regression analysis, testing the variables associated with the serum levels of chemerin. After adjusting for age (β coefficient = 0.028; p = 0.677), disease duration (β coefficient = -0.109; p = 0.107), body mass index (β coefficient = 0.030; p = 0.665), fat mass (%) (β coefficient = 0.072; p = 0.293) and diabetes mellitus (β coefficient = 0.059; p = 0.389), disease activity (DAS28-CRP) (β coefficient = 0.215; p = 0.002) remained associated with serum chemerin levels.

## Discussion

We demonstrated that in women with RA treated with synthetic DMARDs or biologic DMARDs, elevated chemerin concentrations were related to moderate or severe disease activity after adjusting for potential confounders. Serum chemerin concentrations were correlated with DAS28-CRP, swollen joint counts, and CRP, whereas a trend that was not statistically significant was observed between serum chemerin and the number of painful joints.

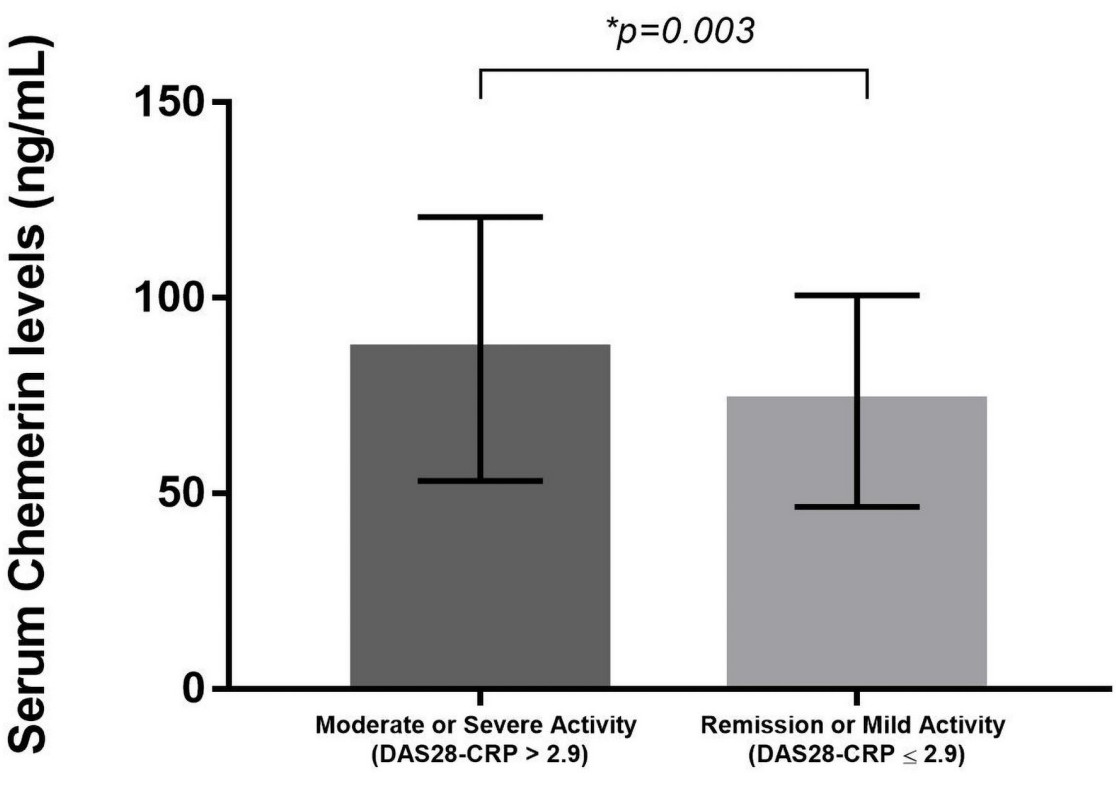

**Fig 2. Comparison of serum chemerin levels between RA with moderate/severe disease activity versus RA in remission/mild activity.**

Some previous studies have investigated the relationship between chemerin and disease activity in RA. Ha et al. investigated 71 patients with RA and found that chemerin levels were correlated with DAS28 and that these levels were higher in active RA [11]. Herenius et al. evaluated 49 subjects with RA treated with adalimumab and observed that chemerin levels were correlated with DAS28 and ESR [12]. Mohammed Ali et al. found that serum chemerin levels

**Table 3. Factors associated with moderate/severe disease activity in patients with rheumatoid arthritis in the multivariate analysis.**

| | Dependent variable: moderate/severe disease activity | | | | | | | | | |
|---|---|---|---|---|---|---|---|---|---|---|
| | B | SE | OR | 95% CI | p-value | B | SE | OR | 95% CI | p-value |
| Age | -0.038 | 0.019 | 0.96 | 0.92–0.99 | **0.047** | NIM | NIM | NIM | NIM | NIM |
| Menopause | 1.088 | 0.518 | 2.96 | 1.07–8.19 | **0.036** | NIM | NIM | NIM | NIM | NIM |
| Disease duration | 0.024 | 0.018 | 1.02 | 0.98–1.06 | 0.173 | NIM | NIM | NIM | NIM | NIM |
| Fat mass % | 0.019 | 0.026 | 1.02 | 0.97–1.07 | 0.460 | NIM | NIM | NIM | NIM | NIM |
| Body Mass Index | -0.038 | 0.048 | 0.96 | 0.87–1.05 | 0.429 | NIM | NIM | NIM | NIM | NIM |
| Diabetes mellitus | 0.862 | 0.440 | 2.36 | 1.00–5.61 | 0.050 | NIM | NIM | NIM | NIM | NIM |
| Chemerin ≥103 ng/mL | 1.077 | 0.382 | 2.93 | 1.39–6.21 | **0.005** | 1.01 | 0.36 | 2.76 | 1.35–5.62 | **0.005** |

Multivariable logistic regression analysis. Dependent variable presence of moderate/severe disease activity. OR: odds ratios; 95% CI: 95% confidence intervals. Crude ORs were obtained using the Enter method. Adjusted ORs were obtained using the Forward stepwise method. NIM: not in the model.

were correlated with DAS28 in their RA patients [18]. The results of these studies support our findings. However, the small sample of RA patients and the presence of confounders that might modify the relation between chemerin with disease activity make necessary to include multivariate analysis, therefore our study also included a logistic regression demonstrating that chemerin concentrations are associated with moderate/severe disease activity.

Chemerin has been proposed as a proinflammatory adipokine in arthritis [24]. Changes in chemerin levels may be related to the inflammatory response in synovial cartilage, which involves chondrocytes, macrophages, dendritic cells, and natural killer cells [25]. In our study, we found a correlation between chemerin and CRP. Similar to our findings, Herenius et al. and Maijer et al. also found a correlation with CRP [12, 26]. Also consistent with these findings, our group previously reported that serum levels of chemerin were correlated with disability function in a small group of patients with RA [15].

Few studies have highlighted the use of chemerin as a predictor of active disease. Tolusso et al. analyzed the utility of chemerin with cutoff values $\geq 95.7$ ng/mL (a cutoff value similar to our study of $\geq 103$ ng/mL). Tolusso observed that their patients with those higher chemerin levels had an increased risk of active disease, and this increase remained after adjusting for other variables [17]. However, our study is the first to identify an increase of 2.76-fold in the risk of moderate/severe disease activity. Previous studies performed in patients without rheumatic disease have reported that patients with diabetes mellitus have higher levels of chemerin [27, 28]. Remarkably, in the present study, we did not observe an increase in chemerin in patients with diabetes mellitus, probably because we did not include patients with uncontrolled diabetes or patients receiving insulin.

Our findings highlight that the inflammatory events driven by chemerin are complex. Chemerin has a chemotactic function for immune cells, promoting cellular migration under inflammatory conditions [29]. Chemerin also participates in activating NF-KB, upregulating the expression of adhesion molecules on endothelial cells, and enhancing monocyte adhesion [30]. Experimental and clinical studies have shown that chemerin production can be stimulated by interleukins (IL), including IL-1β, and it is correlated with CRP, TNF-α and IL-6 in RA [31, 32]. Furthermore, chemerin increases the production of tumor necrosis factor-alpha (TNF-α), IL1-B, and IL-6 by human articular chondrocytes [33]. These previous studies highlighted the relationship between chemerin and proinflammatory molecules.

The present study identified a relationship between high chemerin levels and the severity of disease activity in RA patients; therefore, the results of this work generate new questions about the role of chemerin in the persistence of the inflammatory process in RA. Chemerin is an adipokine synthesized mainly in adipose tissue and liver, and diverse immune cell subsets, including plasmacytoid dendritic cells, macrophages and NK cells, express the chemerin receptor CMKLR1, which when activated promotes chemotaxis and modulates the immune response. In the present study, we identified an association between chemerin levels and CRP and the number of swollen joints; these findings indicate the relevance of chemerin in inflammation in RA.

Nevertheless, our study has some limitations. One of them is that this study was not able to identify changes in chemerin levels. Longitudinal studies are required to demonstrate whether variations in this adipokine might produce changes in other variables in RA. Future studies should be performed to identify whether patients who have persistently elevated chemerin levels might have differences in their therapeutic response to synthetic or biologic DMARDs. Another limitation of this study was the lack of the inclusion of men with RA. We only included women to avoid any possible bias secondary to the hormonal differences that can influence our main variables. Future studies should be conducted on men with RA to identify whether the results observed in the present study persist.

In conclusion, higher chemerin levels increased the risk of moderate and severe disease activity in RA patients. The serum chemerin levels are correlated with higher CRP levels, as well as an increase in the number of swollen joints. This study demonstrates that the association observed between high chemerin levels, and the inflammation assessed by DAS28-CRP was independent of BMI, fat mass and diabetes mellitus, and other potential confounder factors and remains in the multivariate analysis. These findings support the hypothesis that serum chemerin levels could be used as a new biomarker to identify patients with a more severe RA without an adequate response to DMARDs. Nevertheless, future follow-up studies will be required to identify whether elevated chemerin levels defined by our cutoff level can potentially predict the future failure of treatment in patients starting DMARDs. Subsequent studies should evaluate the potential value of maintaining normal chemerin levels as a therapeutic target in these patients.

## Acknowledgments

Members of the Research Group for the Assessment of Prognosis Biomarkers in Autoimmune Disorders.

### Senior researchers

Jorge I. Gamez-Nava, Ernesto G. Cardona-Muñoz, Laura Gonzalez-Lopez, **Department of Physiology, Centro Universitario de Ciencias de la Salud, University of Guadalajara**; Alfredo Celis, **Department of Public Health Sciences, Centro Universitario de Ciencias de la Salud, University of Guadalajara**; Miguel Huerta, Xochitl Trujillo, **Centro Universitario de Investigaciones Biomedicas, University of Colima**; Juan M. Ponce-Guarneros, Maria L. Vazquez-Villegas, **Unidad de Medicina Familiar #4 Guadalajara, Jalisco, y #97, Magdalena, Jalisco, Instituto Mexicano del Seguro Social**.

### Associated researchers

**Research in Clinical and laboratory analyses**: Jessica Murillo-Saich, **Division of Rheumatology, Allergy and Immunology, UC San Diego School of Medicine, La Jolla, CA, USA**; Ana M. Saldaña-Cruz, Norma A. Rodriguez-Jimenez, Melissa Ramirez-Villafaña, **Instituto de Terapeutica Experimental y Clínica, Centro Universitario de Ciencias de la Salud, University of Guadalajara**; Betsabe Contreras-Haro, Alberto D. Rocha-Muñoz, **Departament of Biomedical Sciences, Departament of Health-Disease as an individual Process, and División de Ciencias de la Salud, Centro Universitario de Tonalá, University of Guadalajara**; Cesar A. Nava-Valdivia, **Departament of Microbiology and Patology, Centro Universitario de Ciencias de la Salud, University of Guadalajara**.

### Statistical team

Alfredo Celis, **Department of Public Health Sciences, Centro Universitario de Ciencias de la Salud, University of Guadalajara**; Emilio E. Perez-Guerrero, **Instituto de Investigacion en Ciencias Biomedicas, Centro Universitario de Ciencias de la Salud, University of Guadalajara**.

### Research fellows

Fabiola González-Ponce, Esther N. Sanchez-Rodriguez, Eli E. Gomez-Ramirez, Heriberto Jacobo-Cuevas, Yussef Esparza-Guerrero, Jose J. Gomez-Camarena, Alejandra Martínez-Hernandez, **Instituto de Terapéutica Experimental y Clinica, Departament of Physiology,**

**Centro Universitario de Ciencias de la Salud, University of Guadalajara**; Maria O. Carrillo-Escalante, **Doctorado de Salud Publica, Departament of Public Health Sciences, Centro Universitario de Ciencias de la Salud, University of Guadalajara**.

## Author Contributions

**Conceptualization:** Fabiola Gonzalez-Ponce, Jorge I. Gamez-Nava, Laura Gonzalez-Lopez.

**Data curation:** Ana M. Saldaña-Cruz, Maria L. Vazquez-Villegas, Juan M. Ponce-Guarneros, Miguel Huerta, Xochitl Trujillo, Betsabe Contreras-Haro, Maria O. Carrillo-Escalante, Esther N. Sanchez-Rodriguez, Eli E. Gomez-Ramirez, Cesar A. Nava-Valdivia.

**Formal analysis:** Jorge I. Gamez-Nava, Emilio E. Perez-Guerrero, Alberto D. Rocha-Muñoz.

**Funding acquisition:** Jorge I. Gamez-Nava, Laura Gonzalez-Lopez.

**Investigation:** Ana M. Saldaña-Cruz, Maria L. Vazquez-Villegas, Juan M. Ponce-Guarneros, Miguel Huerta, Xochitl Trujillo, Betsabe Contreras-Haro, Maria O. Carrillo-Escalante, Esther N. Sanchez-Rodriguez, Eli E. Gomez-Ramirez, Cesar A. Nava-Valdivia.

**Methodology:** Fabiola Gonzalez-Ponce, Jorge I. Gamez-Nava, Laura Gonzalez-Lopez.

**Resources:** Ernesto G. Cardona-Muñoz.

**Supervision:** Ernesto G. Cardona-Muñoz, Laura Gonzalez-Lopez.

**Writing – original draft:** Fabiola Gonzalez-Ponce, Jorge I. Gamez-Nava, Emilio E. Perez-Guerrero, Ernesto G. Cardona-Muñoz, Laura Gonzalez-Lopez.

**Writing – review & editing:** Fabiola Gonzalez-Ponce, Jorge I. Gamez-Nava, Emilio E. Perez-Guerrero, Ana M. Saldaña-Cruz, Maria L. Vazquez-Villegas, Juan M. Ponce-Guarneros, Miguel Huerta, Xochitl Trujillo, Betsabe Contreras-Haro, Alberto D. Rocha-Muñoz, Maria O. Carrillo-Escalante, Esther N. Sanchez-Rodriguez, Eli E. Gomez-Ramirez, Cesar A. Nava-Valdivia, Ernesto G. Cardona-Muñoz, Laura Gonzalez-Lopez.

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
