## [Decision Letter · Decision Letter 0]

27 May 2021

PONE-D-21-15276

Serum chemerin levels: a potential biomarker of joint inflammation in women with rheumatoid arthritis

PLOS ONE

Dear Dr. Gonzalez-Lopez,

Thank you for submitting your manuscript to PLOS ONE. After careful consideration, we feel that it has merit but does not fully meet PLOS ONE’s publication criteria as it currently stands. Therefore, we invite you to submit a revised version of the manuscript that addresses the points raised during the review process.

Both reviewers funs some interests in this article, but also pointed out a number of criticisms that require improvement. I ask the authors to fully respond to all comments made by the reviewers in the revised version.. 

We look forward to receiving your revised manuscript.

Kind regards,

Masataka Kuwana, MD, PhD

Academic Editor

PLOS ONE

Journal Requirements:

Reviewers' comments:

Reviewer's Responses to Questions

**Comments to the Author**

1. Is the manuscript technically sound, and do the data support the conclusions?

Reviewer #1: Yes

Reviewer #2: Yes

2. Has the statistical analysis been performed appropriately and rigorously? 

Reviewer #1: No

Reviewer #2: Yes

3. Have the authors made all data underlying the findings in their manuscript fully available?

Reviewer #1: Yes

Reviewer #2: No

4. Is the manuscript presented in an intelligible fashion and written in standard English?

Reviewer #1: Yes

Reviewer #2: Yes

5. Review Comments to the Author

Reviewer #1: 1. Many statements in the INTRODUCTION section contains a scientific fact that need a reference. Please, add a reference after each statement that cite a finding.

2. Exclusion criteria are not sufficient! Do author exclude patients with T2DM, hypertension, CVD,…..etc.

3. Why the means and standard deviation have no decimals? Usually means and SD contain decimals. Also, p-values should contain 3 decimals.

4. In Table 1, p-values should be re-calculated as p-value =1 is not consistent with the results. Other p-values also are incorrect.

5. In Table 2, p-values also wrong and inconsistent with the r-values. I strongly suggest re-stating the results or consulting an expert in statistics because the results are wrong!

6. Please, add the negative error bars in Figure 2.

7. In Table3, Multivariate analysis usually contain Beta(SE) results in the tables. It is not necessary to add “enter method” or stepwise method. Add the decimals to the p-values.

8. Do author make a screening for selection a patients from a specific city or hospital? If yes, Figure 1 is necessary. If not, Figure 1 is not necessary.

Reviewer #2: The Manuscript ID PONE-D-21-15276 entitled "Serum chemerin levels: a potential biomarker of joint inflammation in women with rheumatoid arthritis " can be accepted for publication in PLOS ONE after major revisions.

The purpose of the manuscript is interesting and the obtained results are promising, however, it presents some limitations which needs to be implemented before publication.

The abstract should be re-checked and ameliorated, especially the sentences about the conclusions….which are not the conclusion of the manuscript

The introduction is complete and clear, but the objective of the study should be more impressive…

Concerning method section, the authors should include the description of all the considered demographic and clinical characteristics of the patients enrolled in the study…..as well as the inclusion and exclusion criteria, comorbidities, other pharmacological treatments….

In the results, please describe the flowchart in a better way.

In table 1 please add all the data analyzed.

Please develop the conclusions of the study. The Authors should consider what all of the findings taken together mean, what are the larger implications.

They also need to clarify what advance or advances in knowledge were made by this work.

Check the main text, there are some mistake in English language.

6. PLOS authors have the option to publish the peer review history of their article (what does this mean?). If published, this will include your full peer review and any attached files.

Reviewer #1: No

Reviewer #2: **Yes: **Antonella Fioravanti

---

## [Author Response · Author response to Decision Letter 0]

11 Jul 2021

Guadalajara, Mexico July 09th, 2021

Dr Emily Chenette 

Editor-in-Chief 

Plos One

Ref: Submission ID PONE-D-21-15276

Dear Dr Emily, Chenette: 

We are submitting the new version of the manuscript ID ab5f7e56-9b4a-4376-b0d9-1592d605fd7a entitled: “Serum chemerin levels: a potential biomarker of joint inflammation in women with rheumatoid arthritis”, containing all the modifications that were suggested by the Reviewers’. All the comments and corrections suggested by the Reviewers were considered point by point and the modifications were highlighted in yellow color in this new version of the manuscript.

The questions/corrections suggested by the Reviewers and the answers/modifications to those questions are described as follows: 

 Reviewer 1

1.Reviewers’ comment: Many statements in the INTRODUCTION section contains a scientific fact that need a reference. Please, add a reference after each statement that cite a finding.

Authors’ response: In accordance with the suggestion described above, we have added in the Introduction section (page 5) a reference after each statement that cite a finding. Thank you for this suggestion.

2. Reviewers’ comment: Exclusion criteria are not sufficient! Do author exclude patients with T2DM, hypertension, CVD,…..etc.

Authors’ response: In Methods section (page 7), we included in the study the patients with diabetes mellitus and hypertension. We decided to include these patients with comorbidities and then to analyze in the statistical tests if the chemerin levels were influenced by these comorbidities. In the table 1 we show a comparison of the frequency for these comorbidities. In the exclusion criteria, we have identified that RA patients with other cardiovascular diseases (history of ischemic cardiopathy, myocardial infarction and stroke) were not included. 

3. Reviewers’ comment: Why the means and standard deviation have no decimals? Usually means and SD contain decimals. Also, p-values should contain 3 decimals.

Authors’ response: In accordance with the suggestion, in the Results section (page 13) we now have described in the tables and the text means and standard deviations with their decimals. We have described the p values with 3 decimals. 

4. Reviewers’ comment: In Table 1, p-values should be re-calculated as p-value= 1 is not consistent with the results. Other p-values also are incorrect.

Authors’ response: We have reviewed the p-values. In the Results section (page 14), we substitute the p-value=1 by the term “NC” (not calculated). All the p-values in this new version of the manuscript contain 3 decimals. 

5. Reviewers’ comment: In Table 2, p-values also wrong and inconsistent with the r-values. I strongly suggest re-stating the results or consulting an expert in statistics because the results are wrong!

Authors’ response: We have checked the accuracy of the p values, and these were consistent with the r-values. An expert in statistics was consulted, and the results were corroborated. The p values show the probability that the results were obtained by chance; whereas the r values observed in our correlations denotate the strength of the linear relationship between two quantitative variables assessed. Thank you for your comment.

6. Reviewers’ comment: Please, add the negative error bars in Figure 2.

Authors’ response: We have added the negative error bars in Figure 2. Thank you for the suggestion.

7. Reviewers’ comment: In Table3, Multivariate analysis usually contain Beta (SE) results in the tables. It is not necessary to add “enter method” or stepwise method. Add the decimals to the p-values.

Authors’ response: In the Results section, we have added in the Table 3 (page 17), the values of Beta and Standard Error. We have included in the multivariable analysis presence of diabetes mellitus as comorbid. We have removed of the columns the terms “enter method” or “stepwise method”. We have added the 3 decimals to the p-values. Thank you for these recommendations.

8. Reviewers’ comment: Do author make a screening for selection a patient from a specific city or hospital? If yes, Figure 1 is necessary. If not, Figure 1 is not necessary.

Authors’ response: We have included in the Figure 1, the information of the specific hospital where the RA patients were selected. Thank you for your suggestion.

Reviewer 2

1.Reviewers’ comment: The abstract should be re-checked and ameliorated, especially the sentences about the conclusions…. which are not the conclusion of the manuscript.

Authors’ response: We have rewritten the abstract (page 3 and 4). We have modified the conclusion. Now there is a concordance between the conclusion in the abstract and the conclusion in the text of the manuscript (page 22). 

2. Reviewers’ comment: The introduction is complete and clear, but the objective of the study should be more impressive…

Authors’ response: In the Introduction section (page 6), we have modified the objective of the study according to the comments. Now the new objective is described as follows: “Therefore, the objective of this study was to evaluate whether serum chemerin is an independent biomarker of moderate or severe disease activity in RA patients”. We thank for this important suggestion. 

3. Reviewers’ comment: Concerning method section, the authors should include the description of all the considered demographic and clinical characteristics of the patients enrolled in the study… as well as the inclusion and exclusion criteria, comorbidities, other pharmacological treatments….

Authors’ response: In the Method section (page 7), we have modified the inclusion criteria, to include a more detailed description of the inclusion and exclusion criteria, comorbidities, and pharmacological treatments.

4. Reviewers’ comment: In the results, please describe the flowchart in a better way.

Authors’ response: In the Results section (page 10), we have described in more detail the flowchart. 

5. Reviewers’ comment: In table 1 please add all the data analyzed.

Authors’ response: In the Results section (page 13 and 14), we have added in table 1, all the data analyzed. Thank you for your suggestion.

6. Reviewers’ comment: Please develop the conclusions of the study. The Authors should consider what all of the findings taken together mean, what are the larger implications.

They also need to clarify what advance or advances in knowledge were made by this work.

Authors’ response: We have rewritten the conclusion of the study, in the text (page 22) and in the Abstract section (page 4). 

7. Reviewers’ comment: What all of the findings taken together mean, what are the larger implications.

They also need to clarify what advance or advances in knowledge were made by this work.

Authors’ response: We also added relevant information regarding the interpretation of our findings in the paragraph related with the importance of our study in the Discussion section (page 21), “The present study identified a relationship between high chemerin levels and the severity of disease activity in RA patients; therefore, the results of this work generate new questions about the role of chemerin in the persistence of the inflammatory process in RA. Chemerin is an adipokine synthesized mainly in adipose tissue and liver, and diverse immune cell subsets, including plasmacytoid dendritic cells, macrophages and NK cells, express the chemerin receptor CMKLR1, which when activated promotes chemotaxis and modulates the immune response. In the present study, we identified an association between chemerin levels and CRP and the number of swollen joints; these findings indicate the relevance of chemerin in inflammation in RA”. Thank you for the suggestion. 

8. Reviewers’ comment: Check the main text, there are some mistake in English language.

Authors’ response: The manuscript has been checked and corrected. The English language editing certificate was issued on June 28, 2021 and may be verified on the AJE website using the verification code 216B-E352-B7D8-E106-F506. Thank you for the suggestion. 

Dr Laura Gonzalez-Lopez

Author for correspondence

Primary email: lauraacademicoudg@gmail.com

Alternative email: dralauragonzalez@prodigy.net.mx

Ernesto German Cardona 

Author for correspondence 

Email: cameg1@gmail.com

---

## [Decision Letter · Decision Letter 1]

26 Jul 2021

Serum chemerin levels: a potential biomarker of joint inflammation in women with rheumatoid arthritis

PONE-D-21-15276R1

Dear Dr. Gonzalez-Lopez,

We’re pleased to inform you that your manuscript has been judged scientifically suitable for publication and will be formally accepted for publication once it meets all outstanding technical requirements.

Kind regards,

Masataka Kuwana, MD, PhD

Academic Editor

PLOS ONE

Additional Editor Comments (optional):

Reviewers' comments:

Reviewer's Responses to Questions

**Comments to the Author**

1. If the authors have adequately addressed your comments raised in a previous round of review and you feel that this manuscript is now acceptable for publication, you may indicate that here to bypass the “Comments to the Author” section, enter your conflict of interest statement in the “Confidential to Editor” section, and submit your "Accept" recommendation.

Reviewer #1: All comments have been addressed

Reviewer #2: All comments have been addressed

2. Is the manuscript technically sound, and do the data support the conclusions?

Reviewer #1: Yes

Reviewer #2: Yes

3. Has the statistical analysis been performed appropriately and rigorously? 

Reviewer #1: Yes

Reviewer #2: I Don't Know

4. Have the authors made all data underlying the findings in their manuscript fully available?

Reviewer #1: Yes

Reviewer #2: Yes

5. Is the manuscript presented in an intelligible fashion and written in standard English?

Reviewer #1: Yes

Reviewer #2: Yes

6. Review Comments to the Author

Reviewer #1: All the inquiries have been addressed. I have no further concerns regarding the manuscript. Thank you.

Reviewer #2: The manuscript is well written and organize. The quality of English is good. The paper appears suitable for publication in Plos One journal.

7. PLOS authors have the option to publish the peer review history of their article (what does this mean?). If published, this will include your full peer review and any attached files.

Reviewer #1: No

Reviewer #2: **Yes: **Fioravanti Antonella

---

## [Editor Report · Acceptance letter]

27 Aug 2021

PONE-D-21-15276R1 

Serum chemerin levels: a potential biomarker of joint inflammation in women with rheumatoid arthritis 

Dear Dr. Gonzalez-Lopez:

I'm pleased to inform you that your manuscript has been deemed suitable for publication in PLOS ONE. Congratulations! Your manuscript is now with our production department. 

Kind regards, 

on behalf of

Prof. Masataka Kuwana 

Academic Editor

PLOS ONE